# "Polish People Are Starting to Hate Polish People"—Uncovering Emergent Patterns of Electoral Hostility in Post-Communist Europe

Anne-Sophie Neyra

Electoral Psychology Observatory (EPO), London WC2A 3PH, UK; a.s.neyra@lse.ac.uk

**Abstract:** Like many societies, Poland seems to be increasingly split by the negative feelings many of its citizens feel towards one another because of the ways in which they vote. This phenomenon is known as electoral hostility. This paper sheds light on what it entails in political and psychological terms. A unique feature of this research is its methodological approach, combining family focus groups and individual interviews of up to 70 participants. This enables us to uncover critical insights into the perceptions and experiences of first-time voters and their families. It informs us of Poland's fractious and emotional political atmosphere, but also on the way in which electoral hostility shapes lives in Poland. The findings highlight the importance of mirror perceptions (the perception that others' hatred justifies our own) in shaping electoral hostility as an emotional sequence which makes many voters progressively see their emotions towards opposite voters deteriorate from misunderstanding to frustration, anger, disgust, and ultimately hatred. Finally, the analysis foregrounds the ways in which Polish voters adapt their behavior in accordance with their own preconceived notions of hostility. These preconceptions can manifest themselves via three possible routes: (1) avoidance, (2) aggression, and (3) a sense of doom, deterioration, and hopelessness.

**Keywords:** Poland; electoral psychology; electoral hostility; elections; polarization; behavior





## 1. Introduction

The 2020 presidential election has been the tightest electoral race and the second highest turnout in the history of democratized Poland. The aftermath of the election itself has been equally tense. Several times each week, public demonstrations rallying up to half a million people in single days, have focused on key social and moral questions [1] that have become structural parts of the electoral campaign. These included LGBTQ+ rights and calls for an independent judiciary and free media.

With time, what started as demonstrations against a purportedly authoritarian government progressively turned into patterns of tension and opposition between groups of voters, often involving people who claimed to be neither heavily politicized nor party supporters. In one of its most extreme examples, this resulted in the stabbing of a homosexual couple in February 2021. When asked to react a few days later, Law and Justice (PiS) former minister Witold Waszczykowski suggested that it would be dumb not to assume that there is danger when someone wearing a 'Team B' jersey walks amidst 'Team A' supporters. This phenomenon mirrors what many countries across Europe seem to have experienced in recent years.

This paper suggests that this phenomenon illustrates a critical concept amply discussed in contemporary political science and specifically in electoral psychology: the idea of electoral hostility [2]. Crucially, electoral hostility occurs when citizens' emotions start affecting their attitudes towards each other. This can often result in abstract divisions between opposite camps, but it can also split families, create ill-feelings between friends, and ultimately undermine national unity, compliance, and the very nature of a country's societal fabric–making it uncomfortably fragile. In fact, it often feels as though electoral

hostility may have already virtually permeated most aspects of Polish people's lives—their interpersonal relationships, what they do and do not talk about, the way they consider their similarities, but also their differences with one another.

### 1.1. Research and Operational Questions

This research uses the framework of electoral hostility to shed light on the democratic crisis affecting contemporary Poland. The effect of these tensions on political institutions and parties has been widely addressed in the academic literature, but the critical issue of the extent to which they have reshaped the relationships between and across divided Polish citizens has not been fully unraveled.

This paper answers the following research question:

How have feelings of electoral hostility emerged and developed among Poland's first-time voters and their families?

To do so, four operational questions needed to be addressed:

(1)　What constitutes an 'atmosphere' of electoral hostility?
(2)　What emotional and experiential cycles lead to electoral hostility?
(3)　How do citizens react to electoral hostility?
(4)　To what extent is electoral hostility constructed as a cycle of perpetual attitudinal and behavioral deterioration?

Academic publications may have very different scopes from pure conceptual exploration to the analysis of large systematic data. This article aims for a modest and exploratory, but I feel nevertheless novel and important ambition. It uses qualitative evidence to understand which of two competing theories pertaining to negative electoral tensions within societies–the affective polarization and electoral hostility models, with their conflicting theoretical expectations is best matched by the narratives of 70 Polish first-time voters. First-time voters are a public that attract particular attention in the literature [2]. Specifically, I expect first-time voters' political attitudes to be less tainted by habit than those of more experienced voters. The hope was to receive a narrative that would be more open on questions surrounding perceptions of intra-social relations between voters and less constrained by cumulative habits. The study was conducted without funding. The author acknowledges the exploratory nature of the qualitative approach used, but the number of participants is consistent with qualitative methodologies, and as will be seen, some clear exploratory patterns seem to emerge which can later be confirmed by systematic quantitative analysis. Furthermore, the innovative nature of the approach also relates to the focus on Poland, one of the largest central European democracies, and a party system which has changed a lot more in recent decades than those of the countries most widely studied in the affective polarization and electoral hostility models such as the US and the UK. As such, we hope that the findings of this article provide an important perspective on a theoretical and empirical conundrum.

### 1.2. Approach

Inherently, the question of electoral hostility addresses the articulation between individual and societal dimensions of democratic (dis)harmony. Understanding this will generate insights into how processes of electoral hostility apply to recently democratized societies. Crucially, the Polish case will help arbitrate between two competing theoretical frameworks proposed by research surrounding political behavior and will consequently elucidate similar phenomena.

These frameworks are (1) the affective polarization model, and (2) the electoral hostility model.

This research is built on the conjunction of two qualitative design components: a series of 13 focus groups, and 35 semi-structured individual interviews. In addition, three experts were interviewed to outline the Polish historical and societal context in which the study takes place. I chose three polish academics from different polish cities (Warsaw and

Wrocław), are part of different generations (Gen Z, Millennial, and Gen X), and reside in and out of their country (UK, US, and Poland).

Throughout this analytical journey, a number of key concepts in the fields of political behavior and electoral psychology are addressed, built on, and enriched. These include, but are not limited to socialization, social learning, the political psychology of emotions, and the nature and consequences of democratic hopelessness.

## 2. Analytical Framework

### 2.1. Theoretical Background

2.1.1. Explaining Overlapping Realities with Two Competing Analytical Models: Affective Polarisation (AP) and Electoral Hostility (EH)

This paper sits at the crossroads between critical bodies of literature on political behavior. Some have been at its heart since the 1950s, whilst others have only recently attracted scholarly attention as a result of the gradual transformation of electoral politics.

The arbitration between the affective polarization (AP) and electoral hostility (EH) theories, two major political psychology models, is central to this study. They each seek to explain how and why people hold negative emotions and attitudes towards those who vote differently. It is thus imperative we understand the rationales behind each framework and relate them to adjacent questions such as the role of emotions in politics and in-group and out-group identities. Furthermore, the foundations of this research are supported by vibrant work on socialization and social learning.

The AP model emerged in the US through the works of Iyengar, Sood et al. [3] and Mason [4]. It was built upon and aimed to re-legitimize the classic Michigan model of electoral behavior [5], founded upon the notion that citizens gradually develop a partisan identification, thereby making partisanship a social identity akin to national, ethnic, or religious identities [6]. Consequently, partisanship largely–and consistently–determines citizens' votes, alongside what Campbell et al. [5] label "short-term factors" which encapsulate any contextual element (campaign effects, incumbent's record, personalities, etc.) susceptible to make a voter diverge from their original partisanship. The Michigan model has been heavily criticized not least because electoral choice and partisan identity prove observationally dependent [7]. Thus, in survey parlance, voters "rationalise" their answers on partisanship *based* on how they vote (and redefine partisanship every time their vote changes) rather than voting based upon pre-existing partisan identification. Though these observations have served to marginalize the Michigan model, the AP hypothesis has put it back at the heart of behavioral theories.

AP paradoxically stemmed from increasing criticism towards the classic framework of rational choice theory [8]. Whilst a key tenet of Downs's framework is that in a two-party system such as the US, the median voter will bring about the convergence of political parties, recent literature has shown that reality contradicts this expectation. Despite the US party system remaining bipartisan, parties have ideologically diverged rather than converged of late [9,10]. Consequently, parties and partisans hold increasingly dissimilar (i.e., polarized) positions [1,11]. Furthermore, according to Iyengar, Sood et al. [3], Mason [4], Iyengar and Westwood [12] and Miller and Johnston Conover [13], this increasingly polarized partisanship has led to a new phenomenon known as "affective polarisation", in reference to identity theories whereby increasing in-group identity results in symmetric out-group dislike [14,15]. Put simply, AP suggests that the partisanship model is vindicated by the social identities of radicalized Americans. According to Iyengar and Westwood [12], "affective polarization based on party is just as strong as polarization based on race".

AP gained traction in Western Europe, notably in the works of Reiljan [16] and Hobolt et al. [17] on Brexit. Paradoxically, whilst Hobolt et al. believe Brexit validates AP, it leads Bruter and Harrison [2] to suggest that it may not aptly explain the increasing propensity for citizens to dislike one another because of how they vote. Instead, they argue that the empirical attributes of these negative feelings run counter to the expectations of AP.

First, they note that AP is meant as a partisanship model, and that affective polarization without partisanship is not really affective polarization. This is not merely a technicality but the heart of AP, which relies on partisanship as a social identity and which the literature [14, 15,18,19] systematically finds to be stable over time and rooted in early socialization. Thus, it is not conceivable that Brexit–however impactful–could meet the psychological criteria of a social identity in such immediate ways. In fact, the literature on how social identities adapt to contextual changes [20] explicitly excludes the emergence of such contextual identities.

Second, they emphasize that it is not merely that empirically, neither partisanship nor Brexit might convincingly appear as identity components based on public opinion data. Whilst citizens increasingly express dislike towards opposite voters, the proportion of British, French, or German citizens claiming to identify with a party is at an all-time low, and the proportion of those identifying as 'Remainers' or 'Brexiteers' is well below that of people disliking others based on their Brexit stance. In essence, whilst the affective polarization literature rightly highlights the strength of out-group feelings, it assumes that these out-group attitudes must correspond to a strong in-group identity [21], which is not empirically evidenced. Furthermore, despite political scientists intuitively looking for "positive" identities, the psychology literature has long emphasized the autonomy of out-group hostility, which need not correspond to a strong in-group identity and is instead frequently a symptom of fragile positive identities [22,23] and specifically for aggressive and radical attitudes [24–26]. In addition, Bruter [19] highlights how the dynamics of out-group rather than in-group attitudes rely on a mixture of perceptions, experiences, and mirror perceptions (or indirect reactions).

Third, Bruter and Harrison note that if hostility was to be understood as affective polarization, the most partisan people would be the most negative towards their opponents. Yet, this is not what they find in Europe. Instead, they show that hostile feelings are widespread amongst people who are not partisan (nor "Brexit partisan"), not particularly engaged in politics, and sometimes do not even vote.

Consequently, they bring forward the alternative model of electoral hostility (EH). Stipulating that intellectual genesis is unrelated to partisanship or polarization, they suggest that electoral hostility is a stage of democratic dissatisfaction [27–29] and frustration [30]. They claim that over the past 50 years, citizens of advanced democracies have felt increasingly cynical towards their political elites. Their skepticism has later affected policies and political institutions themselves. They suggest that electoral hostility is the next stage of the democratic dissatisfaction cycle which, after elites, outputs, and institutions, now extends to voters themselves, partly blamed for unsatisfactory democracies. EH is thus different from AP since it:

a.　　is not based on partisanship,
b.　　is an explicit model of out-group negativity (rather than assuming underlying in-group identity behind out-group hatred), and
c.　　primarily affects democratically dissatisfied voters rather than partisans.

This radical shift–and notably conceptualizing hostility as a dynamic stage of dissatisfaction–also leads Bruter and Harrison to use the psychology literature to rethink electoral hostility as an ever-deteriorating cycle of negative emotions. Tying EH to the emotion's literature [31–34], hostility is expected to work at least partly as a Mokken scale (series of increasingly "hard" thresholds) of deteriorating emotions starting from misunderstanding to frustration, anger, contempt, disgust, and ultimately hatred. They also relate electoral hostility to perceptions of a negative atmosphere and expect it to result in an ever-increasing feeling of democratic hopelessness by hostile citizens.

Neither AP nor EH have been fully tested in recent democracies. There have been studies tapping into the concept of polarization in Poland, e.g., [35–37], however, these studies focus on institutional dynamics and the role of parties or the Catholic Church rather than on citizens themselves. Therefore, this study is a unique opportunity to critically contribute to the affective polarization and hostility literatures in two ways: first, by

adapting these theories to a new context in which they remain untested, and second, by using this case study to arbitrate between the two competing models claiming to explain why citizens dislike one another because of how they vote.

2.1.2. Models of Socialization and Social Learning

AP and EH models profoundly relate to issues of identity, emotions, and transmission. The research question in this study explicitly puts family dynamics at its heart. To understand how these dynamics take place, we need to unravel mechanisms of interpersonal transmission, social learning, and socialization within the family and beyond.

Questions of political socialization and transmission have interested political scientists for decades. This literature is shaped by the seminal works of Greenstein [38]; Butler and Stokes [39]; and Niemi and Jennings [40–42]. However, findings on family socialization sometimes appear contradictory. Much of the early literature coincided with the golden age of the partisanship models and primarily looked at whether partisanship is transmitted by parents. Many contemporary scholars have used the European examples of Britain and Belgium to confirm these [43–45]. A considerable subset of recent research also addresses whether family transmission extends to negative partisanship. This is notably the argument made by Boonen [46] in the context of the Flemish extreme right, which confirms similar findings in Northern Ireland [47] and France [48].

Nonetheless, many studies are more nuanced about the political transmission of positive partisanship. For example, Butler and Stokes [39] found that family transmission only occurs if both parents hold homogeneous partisanship and matters less than geographical influence. Greenstein [38] also finds elements of partisan transmission but suggests that they are far less central to socialization than patterns of political debate, such as whether young people learn to freely discuss politics or accept alternative views.

All these findings are directly relevant to this study. Partisan transmission could feed into the shaping of affectively polarized attitudes, and Greenstein's suggestion that attitudes towards political discussion are more effectively compatible with the electoral hostility model could be confirmed, especially as his research shows that politically engaged families and highly partisan families are more likely to encourage critical political debate. This supports the EH expectation that stronger partisanship may not result in stronger hostility but that, instead, families affected by democratic cynicism may be a stronger breeding ground for electorally hostile emotions and their transmission. Similarly, the research on the transmission of negative partisanship could be used to support either model since in EH, negative attitudes and out-group hostility are largely autonomous from positive partisanship, leaving the question of arbitration between the affective partisanship and electoral hostility socialization cycles largely open.

Further complexity pertains to directions of family transmission. Whilst scholars initially looked at top-down vertical transmission from parents to children, these conceptions of socialization have gradually evolved to include horizontal phenomena which view younger generations as active within the process of their own socialization and their consumption of communication flows from family and media alike. This is the argument of Niemi [42], who rejected the conception of the youth as "passive" consumers of the influence of various socialization agents, and McLeod [49] who highlighted the importance of direct and indirect interaction between young people and sources of influence (parents, friends, teachers, or the media) in shaping democratic values and participation.

This "active socialisation" argument has been made specifically in the context of Central and Eastern European democratic transition, notably by Horowitz [50] in the context of Poland, which the author describes as a "natural laboratory [ . . . ] in which political socialization can be studied." He finds that democratic transition and systemic change disrupt the types of generational transmission typically witnessed in the West, and make interpersonal communication far more central to socialization patterns found elsewhere. These findings are confirmed by Koklyagina [51], who highlights the existence of immense generational gaps between the youth and the middle-aged in Poland and Central Europe,

but also by Cammaerts et al. [52] who find differences in youth participation, influence, and the role of interpersonal communication in Hungary and Poland compared to the Western European countries examined (UK, France, Austria, Finland, Spain). Furthermore, Horowitz's [50] work on Poland includes outcomes such as political cynicism and distrust, both of which play a strong role in the electoral hostility model, and which Horowitz finds to be highly correlated with parental education and the nature of political discussion with parents and friends.

Of course, social learning and transmission also occur beyond the family sphere, notably in the contexts of network and contact theories, in both political science and psychology. To understand how political attitudes develop and get transmitted, research has highlighted the importance of interpersonal communication in cognitive mobilization [53] as well as that of the contact hypothesis, showing that recurrent exposure to out-group members reduces hostility [54,55]. More recently, literature on social learning, networks and attitudinal change has been drastically expanded to include the role of social media, e.g., [56], notably in Poland [57]. This has specifically encompassed the risk that the increase in social media usage among Polish and Hungarian citizens may replace exposure to political diversity, and increasingly lead to siloed echo-chambers [58]. Finally, this focus on social learning has been empirically related to questions of political context and atmosphere [59], which also play a critical role in EH.

### 2.2. Theoretical Expectations

Based on the literature on affective polarization, electoral hostility, in-group and out-group identities, the concepts of electoral atmosphere and hopelessness, emotional cycles, and social networks and social learning in Section 2.1 as well as the review of the Polish current historical and political context in Section 3, I derive 14 key theoretical expectations which will be tested throughout this work.

The article tests the suggestion that current electoral tension in Poland follows the patterns of the electoral hostility model rather than affective polarization. It does so by spelling out what we would expect electoral hostility in Poland to reflect using four sets of expectations on atmosphere, perceptions and mirror perceptions, emotional cycle, and consequences of avoidance, aggression, and hopelessness. The specific theoretical expectations are spelt out in Table 1.

**Table 1.** Theoretical Expectations.

| | Analytical Theme | | Theoretical Expectation |
|---|---|---|---|
| 1 | There is a climate of electoral hostility in Poland. | 1.1 | There is a palpable sense of electoral hostility within Polish society; it is acknowledged by different types of voters (regardless of age, partisan preferences, etc.) |
| | | 1.2. | Beyond broader societal spheres, this sense of electoral hostility will progressively penetrate intimate spheres such as families and friendships. |
| | | 1.3. | In a context of changing party system and alignments, I expect the political tension occurring in Poland to fit the theoretical model of EH over that of AP. |
| 2 | Electoral hostility in Poland emerges as a result of hostile perceptions, experiences, and mirror perceptions. | 2.1. | Many citizens will develop hostility towards opposite voters as a result of associating them with negative personal, moral, or psychological characteristics. |
| | | 2.2. | The second source of negative thoughts towards other voters will be either direct or reported experiences of confrontation |
| | | 2.3. | The third route to electoral hostility will come as a result of individuals believing that the other camp displays animosity towards them. This mirror perception will act as a catalyst for hostility. |

**Table 1.** *Cont.*

| | Analytical Theme | | Theoretical Expectation |
|---|---|---|---|
| 3 | Electoral hostility in Poland follows a cycle of increasingly strong negative emotions towards fellow citizens due to their electoral preferences. | 3.1. | The mildest form of electoral hostility will be framed as a lack of understanding towards opposite voters. |
| | | 3.2. | A number of people will express a sense of frustration towards other voters. |
| | | 3.3. | At the next level, individuals may refer to their anger towards those who vote differently from them. |
| | | 3.4. | The fourth emotional stage of electoral hostility will lead voters to use narratives of disgust and/or contempt when talking about those who vote for parties they dislike. |
| | | 3.5. | Finally, the people who have reached the final stage in the emotional cycle of hostility will refer to other voters with a sense of hatred |
| 4 | Electoral hostility in Poland will lead to reactions of avoidance, aggression, and hopelessness. | 4.1. | A number of people will express preferences for avoiding contact with citizens with antagonistic views. Among these preferences, we will find "actively ignoring them" or "avoiding contact and/or political conversations altogether". |
| | | 4.2. | By contrast, many citizens may opt for confrontation and express feelings of aggression towards the opposite camp. |
| | | 4.3. | Finally, in some cases, electoral hostility could lead many Polish citizens to feel a sense of hopelessness, disillusionment, and an irreconcilable feeling of impending doom. |

## 3. Historical and Political Context

It is critical to understand the historical and political context in which the analysis of electoral hostility is situated. While the focus of hostility is on citizens themselves—with their perceptions and attitudes—assessing the local historical and political context at the time of the 2020 Presidential election enables us to sketch the position of political and social elites in context and to understand what historical and political legacies have shaped the current Polish public sphere, political parties, and social systems. The insights from this section are based on expert interviews conducted with three academics and researchers specializing in Polish contemporary politics. Based on their insights, I focused my research on four key elements:

(1) how the country moved from transition to transformation over the past three decades,
(2) the emergence of the currently in place bipolar party system,
(3) key societal and ideological fracture lines and elements of social blame and scapegoating, and
(4) how this has shaped into a specific context under which the 2020 presidential election took place.

### 3.1. From Transition to Transformation

All experts unanimously agreed that it is impossible to understand the climate of 2020 Poland without relating it to the processes of transition that occurred in the aftermath of 1989, and the Great Transformation which followed the initial transition period. The first important contextual element to note is thus that, since 1989, Poland has gone through two successive phases of democratic evolution. Aleks Szczerbiak (AS) defines them as the transition (which he sees as covering the 1989–2005 period) and transformation (since 2005) periods. He also notes that ever since 1989, all Polish elections have been "framed as times for change"—from Communists, from transition elites, from PiS after their first period in power, and finally from more loosely defined liberal elites. Wiktor Babiński (WB)

also confirms that "since 2005 and during the 2015 and 2020 elections, transformation has been a powerful narrative."

According to all experts interviewed, the key difference between the two periods is that while the transition period focused on how various political actors defined themselves vis à vis past communist dictators, the transformation period focused instead on the relationship between current political forces and the transition itself. In many ways, they all point out that the two main actors in the Polish partisan scene—Law and Justice (PiS) and Civic Platform (PO)—are largely defined by their relationship to the heritage of the democratic transition and to the role of Solidarność and the Round Table agreement of 4 April 1989 (between representatives of Solidarność, the rest of the democratic opposition, and the ruling communists led by General Jaruzelski). According to WB, PiS specifically "rejects the founding myth of 1989 as an unqualified success and condemns the Round Table agreements as a rotten compromise." As such, they try to conflate the Communists and parts of the Solidarność elites as aligned and conspiring to the detriment of the Polish people, while claiming for themselves the legacy of the marginal groups which rejected the 1989 transition process and dwindled away on the fringes. This puts them at odds with all other parties on the current Polish political scene. WB also notes that this narrative, which pitches the transformation as being opposed to the transition elites, their strategies, and their preferences, was built retrospectively by the Kaczyński brothers in the 2000s. Indeed, at the time of transition, they had both been personally involved on the side of the pro-Round Table Solidarność elites (Lech Kaczyński was an active advisor of Lech Wałęsa, whilst his twin brother Jarosław was the executive editor of Tygodnik Solidarność, Solidarność's main weekly magazine at the time).

Interestingly, WB argues that both PiS and PO portray themselves in part as the heirs of the Solidarność movement and the enemies of communism, but that "they do so in opposite ways". Whilst PO embrace the heritage of the liberal transition that successfully toppled the Communist regime, PiS portray themselves as the heirs of the more radical fringe of Solidarność which was unhappy with the compromises made with Jaruzelski to ensure a peaceful transition, and who felt that the former oppressors were effectively "allowed to profit from transition". Interestingly, Sławomir Czerwiński (SC) also adds that paradoxically, PiS' "very Manichean" interpretation of transition alternatives meant that "the phrasings and dialectics from the communist period are being reused", albeit in a displaced way. Such a reinterpretation of both the transition and its stakes automatically put PO's "economic modernization credentials" in jeopardy. This transformation focus and conflicting perspectives on the transition period were key factors in placing PiS and PO as the new uncontested leaders of the contemporary Polish bipolar system.

*3.2. The Emergence of a Bipolar Party System*

PiS and the PO only emerged as credible political forces in the early 2000s. As center right parties, they served to occupy the ideological space left vacant by the collapsing historical Solidarity Electoral Action (AWS) and Freedom Union (UW) alongside the more extreme League of Polish Families (LPR) and Self-Defense movement (Zubek in [60]). Gradually, these more extreme movements also fell out of favor, and their electorates became primarily courted by PiS.

As mentioned, the experts interviewed believe that it is this change, from a transition to transformation narrative, which spelt the end of the historical political forces of the end of the 20th century in favor of a PO-PiS duopoly. AS emphasizes how election after election, competition lines shifted and consolidated a "realignment around PiS and PO which started with the 2005 election." He says that this realignment involved a permanent redefinition of what could stand for "change". In 2015, much revolved around the question of whether PO's economic modernization credentials were no longer credible and in 2020 PO was even openly presented by PiS as the defenders of the Third Republic. Conversely, PiS aimed to stand for systemic change (despite having been in power for a significant portion of the period). As part of the process, WB suggests that PiS aimed to "refocus on the argument

of sovereignty, which they accused PO and liberal elites of sacrificing to E.U. integration and cosmopolitan values." Their electoral agenda appealed both to feelings of economic deprivation—often associated with the neoliberal policies of the transition—and the socially conservative resistance to liberal values. It was especially effective at the interjection of those two sentiments. In economic terms, "PiS targeted (among others) those who used to vote for LPR, people who feel that they lost out from the transition, for instance people from small cities where one or two main factories closed down as part of the process of neoliberal economic modernization." They also successfully framed the Smoleńsk air disaster as a major threat to political diversity in Poland and successfully conveyed to voters a sense of symbolic urgency based on the personal tragedy suffered by PiS leaders in the accident.

Conversely, PO also benefited from the political consolidation of the center and liberal camps concerned with the strengthening of PiS as a dominant force of government. AS suggests that this is only one of two components of the current PiS electorate. Noting that PiS lost seven elections between 2007 and 2015, but also managed to systematically reach scores of around 25–30% during that period, he explains that "PiS' winning formula relies on bringing together two different electorates: a socially conservative core electorate, often peripheral and Catholic (including "weekly churchgoers who listen to radio Maryja"), and those who are less well-off and have been disproportionately left out of the benefits of the transition."

WB mentions a similar duality, although framed differently, with (1) a more ideologically radical electorate, still deeply suspicious of the perceived remnants of the communist period and convinced that former supporters of the dictatorship were never fully removed from economic, social, cultural, and political spheres of power, and (2) a more economically revisionist segment of the population who resented being told that the transition was a success when they felt they were losing out from it themselves. Still, according to WB, "PiS was the party that put forward an electorally convincing argument that traced many social ills to the liberal transition being allegedly an unjust process and promised to rectify it. As we shall see, liberal elites have tended to be seen as the first in a series of "scapegoats" at the heart of current Polish political tensions.

### 3.3. Fracture Lines, Social Blame, and Scapegoating

AS, WB, and SC all note that PiS's electoral strategy has followed a populist route of "blaming liberal elites" (AS, SC) for any and all of Poland's current economic, social, and cultural tensions. According to WB, their narrative "contrasts the Polish population at large with the supposed cabal of "former Communists and the liberal transition elites who sold out the interests of the Polish people and personally profiteered from the transition themselves." According to AS, the left-behinds of the transition were particularly receptive to the message because, as time lapsed after the fall of Communism, "a number of older, poorer, more rural voters felt increasingly marginalized" but also idealized the wealth, comfort, and lifestyle of the rest of the country. Indeed, WB notes that the blame narrative was reinforced by many PiS politicians drawing a caricature of "how well ideological liberals from large cosmopolitan cities like Warsaw, Kraków and Gdańsk actually lived and how much they were profiting from the transformation." SC goes further and suggests that the sole aim of scapegoating narratives is to stimulate jealousy towards the progressive middle class.

Whilst social and economic differences are seen as central to fracture lines and scapegoating elements, experts also point to the critical place of the Church, which they feel is comparable to several other Central European nations. AS points out that, despite popular belief, the role of the Catholic Church in Poland has been declining over recent years, and that the perception of this decline has created a sense of backlash from many older, more rural, and less educated voters. He explains that "PiS is seen as the implementer of Church priorities." WB suggests that in this context, "all modern liberal quests" have been framed as cultural threats to the Church, whilst the E.U. has been framed as "a political and economic threat to sovereignty". He notes that these narratives oftentimes overlap,

and that when it comes to scapegoating, the E.U. and cosmopolitan liberal elites tend to merge as a perceived threat to sovereignty and "Polishness", in an escalation culminating in the 2020 Presidential election.

*3.4. The 2020 Presidential Election*

According to WB, what made Poland's 2020 election so specific is that it exemplified PiS' tendency to use major elections as an opportunity to identify and shift its rhetoric towards a new scapegoat. He suggests that in the wake of the European migration crisis, the focus had quite suddenly turned on ethnic minorities and ethnic migrants. However, with the subsequent containment of the crisis, such focus felt obsolete and instead, in 2020, "the fear factor and framing of the election focused on LGBTQ+ rights and sexual freedom". AS emphasizes that the focus on LGBTQ+ rights and abortion was used by PiS to encapsulate the threat to those who see themselves as the defenders of Polish tradition, whilst WB considers that the LGBTQ+ focus was simply a new variation on "liberal elites", the perceived force behind the E.U. in the 2000s, migrants in the 2010s, and current LGBTQ+ and sexual freedom demonstrators.

The Polish election, delayed by COVID-19, marred by controversy over its practical organization in the context of the pandemic, and an initial boycott call in April from three former Polish Presidents and six former Prime Ministers, ultimately took place on 28 June (first round) and 12 July 2020 (second round). At 64.5% and 68.2% in the two rounds, and despite the pandemic, turnout was prominently higher than five years earlier (49% and 55.3% in 2015). In the first round, PiS candidate Andrzej Duda obtained 43.5% of the vote and PO candidate Rafał Trzaskowski, 30.5%.

All other candidates were left well behind (the independent Szymon Hołownia came third with 13.9% of the vote, and all others were below 7%, opening the campaign for a tense second round which Duda ended up winning with 51% of the vote versus 49%. The electoral fracture lines confirmed everything our experts mentioned–a series of gaps between young and old, urban, and rural Poland, Northwest, and Southeast, and along educational and occupational lines.

## 4. Data and Methodology

*4.1. Research Design*

The theoretical expectations that have been developed call for a systematic understanding of how Polish citizens in general, and first-time Polish voters and their families in particular, articulate perceptions, experiences, emotions, and reactions relating to electoral hostility as it occurs within the nation and their very own lives. Achieving this requires a research design that taps into the detailed narratives and dynamics of electoral hostility. This is particularly important since over 90% of behaviors and attitudes relate to subconscious (rather than conscious) sources [2]. Such mechanisms can often be inferred from discursive structures of political attitudes and perceptions. To capture the duality between individual stories and experiences, and the nature of intra-family hostility dynamics, this paper uses a dual qualitative methodology consisting of semi-structured individual interviews paired with family focus groups bringing together two to four members

4.1.1. Family Focus Groups

Family-focus groups were chosen as the first pillar of the research because they enable the researcher to assess dynamics and discussions within a given group. They also enable the comparison of instances of electoral hostility and how they are experienced by different family members. The family focus groups were semi-structured and comprised five parts. These were (1) the description of Poland's current political atmosphere and its perceived determinants, (2) perceptions of opposite voters as well as mirror perceptions (how people feel opposite voters perceive citizens like them), (3) personal and family-felt experiences of electoral hostility, (4) feelings towards opposite voters and current democratic status quo, and (5) consequences of frustration, and expectations concerning the future. Note that in

the context of the focus group protocol, I explicitly made a choice not to ask participants who they voted for as to respect the privacy of their electoral choice. Nonetheless, a large majority of participants ended up voluntarily sharing this during the discussion.

Each focus group comprised between two and four family members, normally including a first-time voter and any of their parents, siblings, cousins, or uncles and aunts. All focus groups were directly conducted by the researcher in either Polish, English or French, based on the participants' preferences. As befits semi-structured methodologies, while the structure of the five parts was generally adhered to (albeit not necessarily their order), the discussion was largely flexible and tailored to the elements dis-cussed by each set of participants. Focus groups lasted 77 min on average. The sanitary re-strictions at the time meant that all focus groups (as well as all interviews) had to be conducted online. They were then recorded, transcribed, and translated.

4.1.2. Semi-Structured Interviews

The second component of the dual qualitative methodology consisted of semi-structured individual interviews. While these do not permit the assessment of interpersonal dynamics, I chose to include them in case individuals were more willing to discuss the depth of their personal thoughts and emotions outside of the presence of family members. Focus groups can sometimes reproduce patterns of heterogeneity between participants, and this is notably true of family focus groups whereby the participants are known to each other and already come with a baggage of roles, habits, and patterns of interpersonal communication. The advantage of individual interviews was thus that they would capture narratives of hostility outside pre-established patterns in case some might be repeated or systematic across families (for instance in terms of differences in internal efficacy or opinion leadership across family members). In addition to family focus groups, I conducted 35 individual interviews. For the reasons discussed above, they were conducted and recorded over Zoom, in either Polish, English or French. I applied the same structure as for the family focus groups, including all five sections, although the phrasing of some of the questions—notably on family habits—was occasionally modified to fit the nature of the individual discussion. Individual interviews were typically a little shorter than the larger family focus groups and ranged from 15 to 90 min with an average of 47 min.

*4.2. Participants and Sample*

In total, I conducted 13 different family focus groups. Family members were either all in the same room or joining the Zoom call separately. The participants ranged in age from 18 to 65. Conversely, when it came to individual interviews, participants ranged in age from 19 to 62. Overall, participants' mean age was 30.2. Overall, I recruited participants from four different Polish areas to ensure sample diversity. These included Warsaw, Wrocław and Poznań. A smaller subset of respondents lived in rural areas from Southern Poland.

Qualitative research does not require full representativeness, and this is not what that selection was intended to achieve, but there are clear differences in political culture and preferences between Warsaw and other large Polish cities, and the countryside. The geographical diversity of the sample intended to avoid a systemic regional or urban bias in recruitment. In total, women represented 44.3% of the total respondents, and a larger proportion of the respondents who declared a partisan preference said that they had voted for PO than for PiS. Participants were recruited using a snowball model. The fieldwork was conducted over two months.

*4.3. Analytical Strategy*

In order to optimize the use of the focus groups and interviews, I devised an analytical strategy aimed at bringing together my analytical and empirical frameworks. I fully transcribed every focus group and every interview. I then created a bank of quotes which followed the thematic components of my theoretical expectations. With the quotes ascribed to each theme, I then proceeded to highlight:

(a)   Unanimous trends that either confirm or contradict my theoretical expectations,
(b)   Omitted trends, claims that none of the participants made even though they could be expected to occur,
(c)   Lines of fracture, elements originating respondent splits,
(d)   Exemplars, quotes which seem highly representative of a trend,
(e)   Outliers, quotes which seem to significantly diverge from the perceptions of other participants, and which thus deserve consideration.

This systematic process, inspired by approaches in Critical Discourse Analysis [61], is the backbone of the qualitative empirical analysis which I develop in Section 5.

*4.4. Ethical Considerations and Conducting Research Amidst a Pandemic*

The fieldwork was conducted between January and March 2021, a time when Poland and Britain were majorly hit by the COVID-19 pandemic. Consequently, and in accordance with the government's response to coronavirus, it was impossible for me to travel to Poland and conduct face-to-face focus groups and interviews, as was my original intent. Instead, and as discussed above, I had to conduct all fieldwork via Zoom. Conducting fieldwork remotely caused challenges to interactions between re-searcher and interviewees, and among participants themselves. To retain traditional dynamics of qualitative research, I asked that all respondents kept their cameras on at all times and took notes recording their tone and any visible and significant body language response, as part of the inter-view/focus group transcript.

Virtual discussions have been the object of intense debate on how ethical consideration should be adapted to ensure optimal data protection at all times. Zoom sessions were password protected to ensure participant privacy. To reinforce ethics protocols, I also ensured that information sheets were sent to participants before fieldwork, and I asked them to sign consent forms. I also made sure I reminded participants of the ethics protocols used in the research at the start of each interview or focus group. Whilst I video-recorded them with their permission, I deleted video content and any identifiable information as soon as I transcribed each discussion. Identifiables were then replaced by anonymous participant codes (unrelated to name and personal information) in all transcripts. These are used throughout the paper.

It is worth bearing in mind that research for this study has not benefited from any funding and is solely exploratory.

## 5. Analysis and Discussion

In my analytical framework, I articulated four sets of theoretical expectations. These were then reflected in the interview and focus group protocols described in Section 4. As a reminder, these four core expectations relate to (1) the climate of hostility within Polish society, and in particular within families, (2) hostile perceptions, experiences, and mirror perceptions, (3) sequences of increasingly grave negative emotions constituting electorally hostile attitudes, and (4) reactions of avoidance, aggression, and hopelessness participants are predicted to articulate in response to electoral hostility. I structure the analysis of participants' narratives along these four key themes using material from both individual interviews and focus groups, and the analytical strategy I described in Sections 4.3 and 5.1.

*5.1. Characterising a Climate of Electoral Hostility within Polish Society and More Intimate Spheres*

In a way, asking individuals to speak of the negative feelings they feel towards fellow citizens—the essence of the concept of electoral hostility—can be an arduous task.

Existing political psychology research has amply shown that people may often feel reluctant to appear negative, critical, or aggressive towards others in the eyes of researchers, creating risks that social desirability may hide the true extent of the phenomena one wished to capture, e.g., [62] To circumvent this difficulty, I chose to start discussions by inviting participants to describe the political climate they experienced every day as

quasi-witnesses. By putting them in a position where they would be both 'insiders' (as Polish citizens in conversation with a foreign interviewer) and 'outsiders' (by describing an atmosphere as witnesses and informers rather than direct actors), my intention was twofold. First, to reduce risks of social desirability and access a more honest description of their perceptions, free of any judgement. Second, to uncover information about the question of "democratic atmosphere" which, as discussed in Section 2, is seen as a critical component of the emergence of electoral hostility in the literature. As mentioned earlier, one of the key components of my analytical strategy consisted in looking for any discursive elements which could have been expected to be present but were not mentioned by anyone. From that point of view, it is remarkable that none of the 70 participants used any positive adjectives whatsoever to characterize the current climate of Polish democratic politics.

Whilst Bruter and Harrison's [2] work on atmosphere suggests that such positive attributes are typically far less prominent than negative ones, to find no such positive reference at all is certainly a striking finding. In fact, from the onset, there is a near-consensus that Poland is going through an unprecedentedly conflictive era, even though the wordings used to characterize this conflict vary in style.

5.1.1. Metaphors of War

There is no more obvious form of hostility than war. Polish people describe a situation which goes beyond that of dissatisfaction or unhappiness—one of outright infighting. In fact, many respondents are so emphatic about it that a significant proportion of the people I interviewed spontaneously referred to metaphors of struggle and war, but also violence and aggression.

This is how respondent 1F220817 felt:

"People are facing each other at a front, like they are at a war [ . . . ] They would be ready to attack each other and go at each other like dogs."

In some cases, war was also directly described as an atmosphere:

"You can feel that tension in the air. This negative aggressive behavior at play is evident when looking back on our Independence Day on November 11th. People set things on fire." (1F220817)

Such bellicose references also extend to descriptions of regular everyday animosity between opposite groups of Poles, as described by respondent 1M221126:

"There have been violent aggressions on both sides, be they physical or verbal. Two opposite extremes have started to emerge." (1M221126)

In their most extreme forms, participant responses included references to the two world wars. We find comparisons to fascism, which remains a particularly loaded reference in the Polish context:

"Poland is an embarrassment. I am really embarrassed when the Polish president speaks. It's a disaster, really. PiS is very close to fascism in a way. That is exactly what it is." (2M440038)

5.1.2. A Tense Atmosphere

In the electoral psychology literature referred to in part two, perceptions of electoral atmosphere have been assessed along two dimensions: positivity, and tension. As a result, not all references to anger and aggression were as dramatic as war.

For instance, respondent 2M230039 preferred to compare the case of Poland to puerile extravagance and immaturity than to open war.

"Poland is an angry child." (2M230039)

"People insult one another, it becomes very vulgar." (1M221126)

Tension is also referred to in more benign ways:

"It's insults everywhere. Left-wing guys calling right-wing guys xxx or right-wing guys calling left wing guys xxx." (1M221126)

"Relationships are more heated than before." (1F220817)

Finally, descriptions of the atmosphere of Polish politics as of 2021 explicitly reference electoral hostility itself:

"Now it's two segments of the population fighting one another." (1M221126)

"Now, the fault line goes across society." (2M44038)

"The political scene is divided. There is society and then, there is 'the others'." (2M440038)

"Generally speaking, yes, people are split. This means that there is a lot of hate, physically and over social media." (1M181230)

Note that, crucially, when it comes to our theoretical model, many participants used references which support hostility over polarization by sending voters back-to-back:

"If it was PO, I would feel the same." (2M440038)

"Me? I'm a snob, I think they are all wrong!" (2M260053)

### 5.1.3. Hostility within Family and Intimate Spheres

Electoral hostility is also referred to in the context of intimate circles—within families and between friends. There were countless references to the excruciating atmosphere surrounding Christmas and other family gatherings:

"When it comes to families, they are trying to avoid meeting." (1M221126)

"Many family relationships in Poland have totally collapsed." (2M55045)

"Whenever we have family meetings, we have to keep politics out of it." (2F230040)

References to a tense atmosphere also occur–albeit a bit less frequently–in the context of friends and partners:

"Especially in making new friendships. The first thing you are going to wonder is what political party they vote for." (2M220056)

Ultimately, the atmosphere of both Polish politics and intimate circles is seen as poor, tense, and unhealthy. As mentioned at the start of the section, there is virtually no positive reference made to either whatsoever.

### 5.2. *Perceptions, Experiences, and Mirror Perceptions of Hostility*

One of the key challenges of understanding the dynamics of electoral hostility rests in disentangling the roles played by perceptions of opposite voters, direct experiences of hostility, and mirror perceptions (the perception that it is the opposite camp which holds negative or discriminatory attitudes towards us). The complex relationship between direct and mirror perceptions has long been a critical part of the psychological and political science literatures on identity as well as on prejudice and discrimination (notably on racism and antisemitism). In that sense, it is unsurprising that this relationship should play an equally significant role in the literature on electoral hostility which is, in a sense, a form of prejudice.

### 5.2.1. Perceptions of Opposite Voters

Both EH and AP rely on individuals perceiving opposite voters as a largely unified group and describing it as such. There is, however, an important difference between the two models. In AP, partisanship is defined as the 'primary identity' (see Section 2) and as such, there is no need for individuals to resort to other 'lower' elements of descriptive characterization. This is because psychological research on in-group and out-group identities would suggest that, when a primary basis of identity exists, individuals prefer to focus on

it rather than on secondary (and therefore less relevant) bases, e.g., [23]. AP would also lead individuals to see themselves as another equally homogeneous group of partisans based on their own positive partisan identity.

In contrast, EH is not based on partisan identity, and this results in two differences. First, individuals will not see themselves as defined by their own partisanship nor necessarily as part of a homogenous partisan group. Second, while the derogative characterization of other voters might occur (in the words of Bruter and Harrison [2], due to "incompatible moral hierarchization", 309), individuals will additionally need pseudo-objective characterizations of opposite voters in quasi-cleavage terms, that is, by engaging in over-simplifications of the social, demographic, or cultural characteristics of the group on the receiving end of their hostility.

On these grounds too, the interview material overwhelmingly confirms the expectations of EH over AP. Very few respondents would refer to themselves as "PiS voters" or "PO voters" and do not seem inclined to describe the group of voters which they themselves belong to. In fact, many explicitly reject positive partisanship and stress that they would characterize voters for parties which they support just as negatively if their tone or action justified it. Second, when it comes to opposite voters, they largely embrace two layers of descriptive and judgmental characterization, thereby seemingly contradicting the primary positive identity of AP.

Social, Demographic, and Cultural Characterization of Opposite Voters

Attempts to characterize the nature of opposite voters on the basis of 'objective' characteristics involve a mixture of social, demographic, and cultural characterizations which primarily involve age, geography, education, religion, and, strictly from the point of view of PiS voters, sexual orientation. Some of these characterizations are illustrated in Table 2.

**Table 2.** Social, demographic, and cultural characterization of opposite voters.

| Age | Geography | Religion | Sexual Orientation |
|---|---|---|---|
| "There is also a generational divide." (2M21004) | "There are entire districts where people are stupid." (1M221126) | "She listens to Radio Maryja!" (2M220041) | "An old man shouted at them-LGBT!" (1M181230) |
| "With young people, I talk to them, I share my views with them." (1F550511) | [of a rural village] "There was a Trzaskowski poster. The next morning, it was covered in manure." | | |

From Derogatory Comments to Insults

Beyond descriptive comments, plenty of derogative characterization of opposite voters, sometimes to the point of insults, were recorded. These are some examples:

"They are troglodytes!" (2M550045)

"I don't mean to be rude, but they don't dress well." (2M230047)

"They are thugs!" (1M221126)

"The traitors!" (2M440038)

"Are swear words included on that list? [ . . . ] I'd call them 'fuckers'!" (2F230040)

"They are aggressive and act like animals because they have half-brains." (1F220714)

"I don't know where they come from. Maybe from the moon, maybe from Mars, maybe from a psychiatric hospital." (1F550613)

"I don't feel threatened because they are stupid, and I don't think that anybody who is stupid can threaten me." (2M440038)

### 5.2.2. Experiencing Electoral Hostility

Electoral hostility is not merely about perceptions, but also about experiences—and part of the question is whether hostility is mostly 'imagined' or 'experienced', be it societally or within intimate circles.

Experiences of Societal Hostility

Fieldwork participants were split between those who had not experienced electoral hostility in their social life, and those who did. Overall, only a minority of participants had experienced hostile speech or actions directly from opposite voters that were not part of their personal circles. These memories were usually stark and sometimes upsetting:

"We got into political fights, so many times." (2F230040)

"They will bully me into believing that they are right." (2M440038)

"Literally, she insulted me." (2F220036)

"He kind of attacked me right away." (2M440038)

"My husband was at a dinner where he disagreed with someone on an issue [ . . . ]. He got up, and my husband was convinced that he was going to hit him." (1F550511)

"Yes, we always fight, viciously, every time." (1M480307)

Experiences of Electoral Hostility within Families: An Awkward Uncle

Whilst only a minority of participants had direct experiences of hostility from strangers, these experiences were more frequent when coming from family members, friends, and acquaintances.

Experiences of electoral hostility within families are often rare within the immediate nuclear circle, but rife beyond it and were abundantly and vividly described:

"I have friends who don't meet up at Christmas because of political divides." (1F550511)

"My uncle on my dad's side votes for PiS [ . . . ]. I think it's harder for my dad [ . . . ] because he does not have a relationship with his brother anymore." (2M2200326)

"It's only with my brothers and sisters with whom we can have more or less civilized debates." (1M221126)

"At the dinner table, between the drunk uncle and the grandmother, there is always going to be trouble." (2M2200329)

"I have a funny uncle." (2F220064)

"Everybody has a funny uncle!" (2F220048)

### 5.2.3. Mirror Perceptions

Finally, mirror perceptions occur when people believe that it is not *them* who dislike opposite voters, but rather that opposite voters dislike them to such a point as to make reciprocal hostility effectively unavoidable. References to mirror perceptions were in fact endemic in both family focus groups and individual interviews. Whatever participants would say they felt towards opposite voters, they were almost universally convinced that opposite voters lived these emotions more intensely.

Mirror perceptions are at the heart of EH because of its rooting in psychological theories of prejudice and discrimination and because such perceptions disinhibit one's own hostile feelings and behavior without making them feel any guilt. Here are a few examples of how they are framed:

"I don't have an enemy, but I certainly am theirs." (2M440038)

"They hate us!" (1F220714)

"So clearly, I'm a threat, an abomination, and an insult to God." (2F230031)

"They would probably not like me either." (1F220102229)

"Please, do not make devils out of people who have different views than leftists." (1M220037)

"He even felt disgusted by me I think." (2M440038)

"I think PO voters treat PiS voters like they're a joke, like they are not people with whom they can have a sensible, reasonable conversation with." (2M440038)

The disinhibiting effect of mirror perceptions is also obvious in some cases:

"I attack them, but because they attack me." (2M440038)

Finally, in one specific case, a participant summarized the paradox of mirror perceptions and perceptions as being apparently unbalanced. They admitted that ultimately, both are probably equivalent:

"PiS voters see us probably the same way as we see them." (1M181230)

Electoral hostility patterns are confirmed as a triangle of systematized perceptions of opposite voters (both pseudo-objective and openly derogative), direct experiences of hostile speech or behaviors (mostly from family members and close ones with diverging opinions), and mirror perceptions which disinhibit hostility and aggressiveness as part of a "they started it first" argument.

The process seems to favor expectations related to EH over AP. This is noteworthy because individuals do not seem to see themselves as part of a group of like-minded voters (indeed, an overwhelming majority refer to those who vote for the same party as them in the third person rather than the first person), sometimes even leading to an explicit choice to disassociate from like-minded voters. It is also apparent in the fact that partisanship does not seem to work as a primary identity, causing respondents to characterize opposite voters in pseudo-objective demographic, social, or cultural terms instead.

At the same time, this triangle unleashes powerful emotions, which we now explore.

*5.3. An Ever-Deteriorating Cycle of Hostile Emotions*

Electoral hostility is presented as a sequence of progressively worsening emotions that a hostile citizen will feel towards opposite voters, starting from a first stage of misunderstanding to frustration, anger, contempt, or disgust (presented as two alternative mobilizing and demobilizing routes), and ultimately hatred. In this section, I assess whether participant narratives mirror these emotional categories.

The most benign stage, that of misunderstanding, is well expressed by participants who talk about opposite voters as being hard to understand or illogical in their behavior. This is for instance exemplified by participant 1F220102229:

"I don't understand what their thought process is."

References to frustration are even more frequent. It is either sometimes referred to as a sense of tiredness or powerlessness when reasoning with opposite voters, who are often portrayed as obtuse or entirely closed to any form of discussion or compromise. However, more often than not, frustration is simply openly expressed. For instance:

"I am frustrated and really hopeless." (2F620065)

"It's so frustrating!" (2M440038)

References to anger are almost as frequent, and once again, they tend to be very direct, with participants describing their own anger, their willingness to scream and shout–or their own worry that they will. See the following:

"I am pissed off; internally I am screaming!" (2F220036)

"It simply irks me." (1M221126)

"If I saw a beautiful girl with a Duda 2020 t-shirt, I would not even feel sad but angry. I would be pissed off that she's in the same room as me!" (2M20035)

For another set of participants, their own anger is an emotion which surprises them, one which they did not expect to have in themselves, often witnessed with surprise, as though electoral hostility was unleashing a part of them which they do not usually see:

"I was being very sarcastic, and mean, and aggressive because I was angry." (2F230031)

"It's emotional. We do fight and argue." (1M480307)

The next stage in the electoral hostility emotional cycle involves either contempt or disgust, depending on whether individuals are more prone to mobilizing or demobilizing emotions.

Both types of emotions proved meaningful in the discussion, and, interestingly, it is also the case that very few participants would express both disgust and contempt at the same time. As such, they effectively work as two possible–and seemingly emotionally exclusive–routes: anger and hatred.

Contempt usually takes the form of derogative references to opposite voters' thought processes, such as deeming them illogical, stupid, or even dismissive. For instance:

"Even under communism, they weren't that primitive." (2M440038)

"They are troglodytes!" (2M550045)

"Yeah? Well, they can say whatever they want!" [said with a dismissive tone] (1M221126)

Disgust is usually expressed in an even more direct and open way, in reference to an overwhelming and almost physical sense of rejection towards opposite voters and a perception of one's own incapacity to deal with them:

"I would be disgusted by them." (2F220064)

"A couple days later, my friend told me that one of the guys we hung out with at poker night was a PiS supporter. Something tilted in me, I was like ugh, eww, ehh, I'd probably have a yikes moment if this happened again." (1M231228)

"No, don't touch me. I feel disgusted!" (2F220064)

"I have disgust and dislike towards them, not because of their political views but because of what I think they are as human beings." (1M231228)

Finally, the ultimate stage of electoral hostility is that of hatred. Once more, hate often tends to be mentioned very openly, almost candidly, by several participants. At times, whilst some identify their own feelings, others make use of hateful references to describe the opposite side, and finally, some will simply describe behavioral intentions (real or fantasized) that seem to proceed from hatred. Here are examples of each of these three models of reaction:

5.3.1. Direct Identification of Hateful Feelings

"I hate them, I hate the party, there is no party that I hate more!" (1M181230)

5.3.2. Hateful References

"Kaczyński is the devil!" (1M181230)

(In this case fantasized) behavioral intentions proceeding from hatred)

"I want to rip their heads off!" (2F220036)

To conclude this section, beyond confirming that each of the expected emotions is represented in the discourse of at least some participants, I wanted to assess their frequency. I looked through each participant's transcripts, searching for words corresponding to each of these emotions (for instance "frustrated" or "fed up" for instances of frustration, or "stupid" and "uneducated", for instances of contempt).

As a reminder, since the sample was not designed to be representative, it is not possible to derive specific proportions of respondents expressing each of these emotions, but it is still possible to differentiate broader categories of occurrences, which I describe as anecdotal, occasional, regular, frequent, and endemic (corresponding to approximately up to 5%, 10%, 20%, 50%, and over 50% of participants using words corresponding to each emotion). The results are mapped in Table 3. It shows that frustration is the type of emotion occurring most frequently in participants' comments. Frustration is then followed by anger, while disgust and contempt are the most infrequent types of emotional references.

**Table 3.** Emotional cycle and occurrences.

| | | | |
|---|---|---|---|
| Endemic, 25+ participants (>50%) | Frustration | | |
| Frequent, 15–30 participants (<50%) | | Anger | |
| Regular, 8–14 participants (<20%) | Misunderstanding | | Hatred |
| Occasional, 4–7 participants (<10%) | | Disgust | |
| Anecdotal, 0–3 participants | | | Contempt |

Furthermore, because of the Mokken scale nature of EH, one would expect "easy" forms of negative emotions (such as misunderstanding and frustration) to occur more frequently than 'hard' ones (such as disgust, contempt, and hatred). Broadly speaking, this is the pattern found.

While the relationship between cycle place and occurrence is not exactly linear, this may largely be explained by the fact that citizens may ultimately move through stages of hostility, meaning that by the time they express disgust or hatred towards those who vote differently from them, it is likely that they no longer refer to their sense of misunderstanding towards them. In other words, the cycle of hostility is dynamic and as such, each new stage seems to result in a new set of dominant emotions rather than simply being additive.

*5.4. Consequences of Electoral Hostility from Avoidance to Aggression and Hopelessness*

Ultimately, while it is important to know whether electoral hostility has emerged in contemporary Poland, and whether EH is a more convincing model to explain current tensions and negative feelings between Polish people than AP, it is critical to understand what the consequences of such hostility may be. Three key attitudinal consequences are of interest here: avoidance, aggression, and hopelessness.

Avoidance works as a form of withdrawal, whereby citizens may be so reluctant to engage with hostility (or in some cases, be so traumatized by it), that they prefer to avoid confrontation with diverging citizens, either accepting to meet with them but on the condition of not talking about politics, or even avoiding them entirely, thereby reinforcing an ideological harmonization of one's social circle and ensuing silo effects.

By contrast, aggression suggests that electoral hostility gets formalized into patterns of open ideological and social contention with a citizen choosing to stand up to the other camp and engage in open and often regular confrontation.

Finally, hopelessness is a critical attitude in recent electoral psychology literature. This reaction pertains to a sense of doom and certainty regarding future deliquescence and deterioration which seem inevitable.

Avoiding conflict?

Political psychology research suggests that many people are "conflict-avoiding" [63]. As such, one of the predictable reactions to a sense of electoral hostility will be, for many,

to opt for avoidance as a form of self-protection. In narratives, this avoidance takes three different forms. From mildest to more radical: avoiding talking about politics with people one disagrees with, avoiding people one disagrees with altogether, or the temptation to leave the country.

Many participants went to great lengths to explain how they aim to avoid political discussions with random strangers, or even with close friends and members of their own families. A few illustrations of these strategies are presented below:

> "We don't talk about it. The topic just doesn't come up." (1F450306) "The topic just doesn't exist." (1M480307)

> "If I hear people talking about politics at a party, I just leave the conversation. I don't feel like talking about it [ . . . ] so avoid these conversations." (1F220714).

At times, participants try to make fun of such a desire for avoidance:

> "Everyone avoids it because you don't want to get into a war, hahaha." (2M440038)

Others differentiate between people they can avoid altogether and those whom they have to see and with whom they have to avoid political conversations instead:

> "With friends, I guess it's easier because you can dissociate yourself from friends more easily; whereas family, they're just there." (2F240055)

> "I would be more eager to see my cousin if he didn't have the views he currently has." (2F230031)

This leads to the second form of avoidance, whereby a number of participants choose to ignore people known to them to be opposite voters. In short, they simply avoid potentially explosive conversations.

This is expressed in a number of different ways by different voters—peaceful or angry, on social media, or in real life.

Participant 1M221126 is a good example of social media avoidance:

> "There are some people I've stopped following without blocking them as such."

By contrast, several participants are very candid about their choice to peacefully stick to people whose political ideas are compatible with theirs:

> "I avoid them. I don't want to speak to them. I choose not to speak to them." (2M440038)

> "I don't want these people. I am very selective about my entourage [ . . . ] and I don't want these people around me." (2F220036)

> "I'm done arguing. [ . . . ] I prefer to ignore them." (2M440038)

At times, however, coming to the conclusion that one *must* avoid opposite voters is framed as a more heated–or sometimes aggressive–decision, almost a form of punishment:

> "I don't feel like sharing anything other than pavement and street with them." (1M240818)

> "I cannot speak to these people. [ . . . ] I cannot stay in the same room as them." (2M440038)

The third and final level of avoidance is the temptation to leave Poland altogether, that is, not simply refocusing conversations or readapting one's social circle, but considering that Polish society as a whole has become unlivable due to the ambient electoral hostility, as shown by the below:

> "One of the main triggers for me to leave Poland." (1M181230)

> "Haha, well, I guess now it's time to emigrate!" (2M220032)

> "I actually know people who moved away because they were just scared." (2F220048)

Aggression–the choice of fighting back?

While many participants suggested that their preferred way to react to hostility is avoidance, several others prefer to openly embrace conflict with opposite voters. This feeling is articulated either as a reaction to opposite voters, or even, sometimes, as a reaction to close ones (such as family members) developing strong ties with opposite voters:

"They can all f*** off. I can't be bothered!" (1M221126)

"Marry someone from PiS? Jesus!" (2M440038)

"If my younger sister dated a PiS voter, I would have an intervention, I would tell her that she cannot date him anymore!" (2M0035)

On balance, however, it is worth noting that references to aggression were far less frequent than references to avoidance. At times, they also seemed to betray a hint of the third type of reaction to electoral hostility that many, in fact, most, participants felt: that of hopelessness.

Hopelessness

As discussed in Section 2, hopelessness is the quintessential attitudinal consequence of electoral hostility and, in many ways, one of the key differentiators between EH and AP.

Affective polarization does not lead to hopelessness because, however unhappy partisans might be when their party loses an election, they will also believe that the world is functioning well when their party wins instead.

By contrast, in EH, hostility is seen as the final stage of the progressive deterioration of perceptions with regard to the political system itself and, as such, those who experience such hostile feelings can be increasingly devastated by the negativity it entails, regardless of electoral outcomes. This is also due to the core difference of EH, which stems from a self-standing negative identity, whereas AP only sees this negativity as the symmetric derivative of the positive (in-group) identity that precedes.

From that point of view, it is noteworthy that references to hopelessness were, in fact, endemic in both interviews and focus groups. Respondents expressed sadness, predicted a future marked by the continued deterioration of the country's democratic atmosphere and hostility, and ultimately could not imagine a route for improvement. Here are a few illustrations of how feelings of hopelessness were expressed, starting with the narrative of sadness:

"It's sad, but that's the way it is." (1M221126)

"So now, she feels particularly emotional about things. When PiS-related news come on, she will cry, she will scream." (2F230031)

"It was hard, it's been hard for a long time. It cost me certain mental health problems such as depression." (2M2200326)

"It is just depressing for us." (2F220036)

"You are attacked by these things, every week, constantly." (2M440038)

A more acute extension of a similar feeling is found in a narrative of despair:

"I feel sad, disgusted, flabbergasted." (2M2200326)

"I am really hopeless." (2F620065)

"Despair. That's what it is." (2M550045)

"I am a foreigner in my own land. These are very strong feelings, again." (2M2200326)

Finally, the last symptom of hopelessness is encapsulated by the perception of doom and expected continuous deterioration. This is another set of references that appeared very regularly in both interviews and focus groups:

"Is [electoral hostility] an increasing reality in Poland? Absolutely!" (2M230039)

"It will certainly deteriorate." (2M55045)

"I am afraid things might get even worse. Things will get pretty bad." (2M440038)

"Things will get worse, absolutely. I think we have not seen about half of what's to come. We are in for some really nasty surprises." (2M220033)

"The difference between attacking somebody's views and somebody as a group, is not present in Poland anymore. This is very dangerous." (2M210042)

"Political aggression is certainly accelerating." (2M10042) "I don't know where we are going anymore." (2F220036)

"There is no emergency exit." (1F551024)

The hopelessness and never-ending negative spiral argument is perhaps the strongest analytical argument of all, suggesting that the situation observed in 2020 Poland fits EH more than AP.

PiS supporters were not any more optimistic about the future than were PiS opponents. The overarching perception across left and right-wing interviewees, be they politically engaged or completely detached, was largely similar–not only is the situation bad but it is getting worse.

Altogether, one can observe a triptych of gradual and increasingly serious consequences resulting from electoral hostility. These range from four different stages of avoidance to aggression and ultimately hopelessness, all of which can be combined. This triptych is encapsulated in Figure 1.

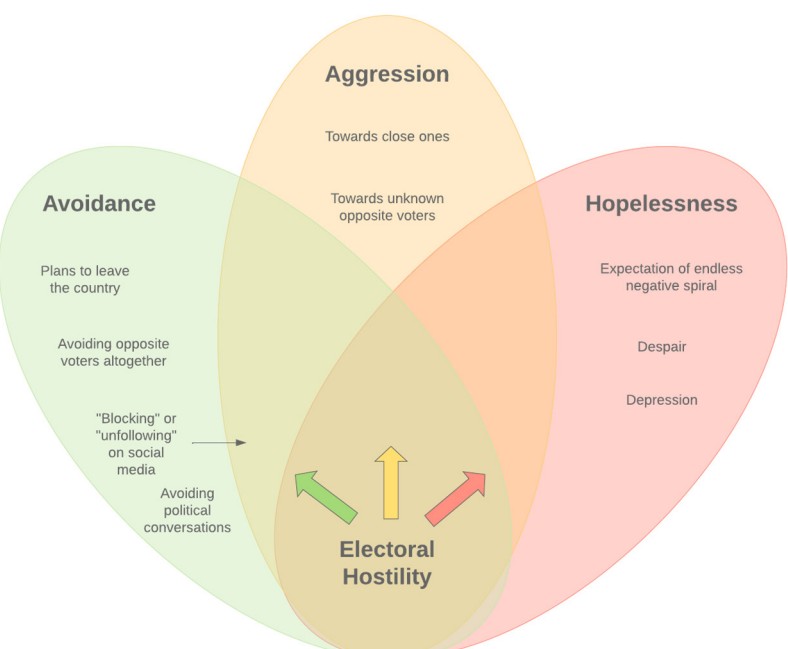

**Figure 1.** Consequences of hostility.

The difficulty in arbitrating between affective polarization and electoral hostility is that as pointed out by Bruter and Harrison [2], they converge on the central prediction that people will develop negative emotions towards voters for opposite parties. Yet, the electoral hostility model also points out that it differs from the affective polarization model in several important expectations (or, in some cases, implications). Throughout the analysis, I have focused on these explicit divergences to arbitrate between the two models and assess whether, as suggested by proponents of electoral hostility, there is indeed a strong case for their model. In Figure 2, I summarize the outcome of the "comparative test" of the affective polarization and electoral hostility models, and the conclusion is that indeed, in the Polish case, the prima facie findings strongly support the electoral hostility perspective.

First, unlike affective polarization theory, the electoral hostility model focuses a lot on the importance of electoral atmosphere. References to it are plentiful in my exploration. Second, in the affective polarization model, out-group exclusion is the corollary of a strong in-group identity, whilst the electoral hostility model suggests that out-group negative emotions do not require in group identification. In the case of Polish first-time voters, references to in-group identity were indeed extremely scarce. Unlike electoral hostility, the affective polarization model necessarily focuses on partisanship as primary identity, and again, partisanship is not seen as the critical foundation of electoral hatred by Polish first-time voters. The electoral hostility model focuses on the importance of mirror perceptions which are prominent in the interviews and focus groups. It also describes a specific cycle of ever-deteriorating emotions, all of which are symptomatically mentioned in Polish first-time voters' narratives. The electoral hostility model also suggests that they are organized like a quasi-Mokken scale of increasing gravity and that citizens will move through them in stages, and again, both components seem verified and accounted for by the interviews and focus groups. Silo effects are compatible with both theories and indeed heavily present in the evidence. However, finally, the two models diverge strongly on the im-plications of having one's party in power and of partisanship strength. The affective polarization model suggests that strong partisans will be more likely to show symptoms of aggression and that having one's party in power will prevent symptoms of hopelessness. By contrast, the electoral hostility model suggests the opposite on both counts, with strong partisans being less likely to indulge in symptoms of aggression (since the electoral hostility pattern is seen as a continuation of cynicism phenomena) and having one's party in power not serving as an antidote to hopelessness. On both fronts, my findings strongly confirm the electoral hostility approach.

| | **Affective Polarisation** | **Electoral Hostility** | AP | EH |
|---|---|---|---|---|
| **Atmosphere** | - | Atmosphere of hostility at the heart of the model | N/A | ✓ |
| **Perceptions** | Reference to ingroup identity | No reference to in-group identity | ✗ | ✓ |
| | Primary outgroup identity is partisanship | Will require out-group pseudo-objective characterisation | ✗ | ✓ |
| | Ingroup identity leads to polarisation | Mirror perceptions is used to justify hostility | ✗ | ✓ |
| **Emotions** | - | Deteriorating cycle of emotions | N/A | ✓ |
| | - | Works as a partial Mokken scale | N/A | ✓ |
| | | Works as cycle so citizens move through "stages" of emotions | N/A | ✓ |
| **Consequences** | Silo effect | Avoidance is key | ✓ | ✓ |
| | Aggression is more likely as people become more partisan | Aggression is less likely | ✗ | ✓ |
| | No hopelessness if in power. Hope of return to power otherwise | Hopelessness critical whether party won or lost elections | ✗ | ✓ |

**Figure 2.** Summary: AP vs. EH.

## 6. Conclusions

Throughout this dissertation, I aimed to understand how feelings of electoral hostility have emerged and developed among Polish voters, particularly among first-time voters and their families. I did so using qualitative fieldwork involving 70 participants (including 13 family focus groups with 35 participants and 36 individual interviews, in addition to 3 expert interviews).

Key findings are summarized in Figure 2.

Firstly, there is an overwhelming sense that there is a generalized atmosphere of hostility not only within Polish society as a whole, but also within more intimate spheres. Not a single participant used positive adjectives to qualify this atmosphere.

Secondly, electoral hostility has shown to emerge from the interaction between (largely negative) perceptions of opposite voters, direct experiences of hostile speech and/or action (only a minority experience it from strangers, but many deal with it within intimate circles), and mirror perceptions, whereby most participants are persuaded that others hate them more than they hate others themselves. This is important because the three sides of the triangle consistently reinforce one another while the mirror perception component in particular is disinhibitory, thereby enabling people to hate fellow citizens while persuading themselves that it is, initially, the other person's fault.

Thirdly, electoral hostility has proven to develop as a cycle of ever-worsening emotions, starting with misunderstanding, continuing on to frustration, anger, disgust, contempt, and ultimately hatred. Frustration is most frequent, and disgust and contempt are least frequent. The cycle acts as a partial Mokken scale but individuals also move along stages as their hostility strengthens, partly replacing the mild emotions of the beginning by significantly stronger ones.

The consequences of electoral hostility are threefold.

(1) avoidance (which can mean avoiding talking politics with disagreeable people, avoiding disagreeable people altogether (on social media or in real life), or even being tempted to leave the country),
(2) aggression, and
(3) hopelessness

Aggression is the least common outcome. By contrast, hopelessness, which may include depressed feelings related to politics—but also a sense of doom and of an ever-worsening spiral—is far more widespread than would have been expected.

One of the key analytical contributions of this paper has also been to arbitrate between two alternative theoretical models, namely affective polarization (AP) and electoral hostility (EH), equally predicting that citizens will hold negative attitudes towards one another.

Throughout the analysis and discussion of the findings, I have highlighted areas in which the two models led to diverging expectations. I summarize these elements in Figure 2.

Ultimately, the results overwhelmingly support EH. This is particularly so because hostility does not seem to rely on positive in-group identity (partisanship) given that it affects 'winners' just as much as 'losers' of the election, and because all the psychological expectations relating to the emotions and reactions of hostile citizens effectively seem confirmed by the Polish case.

This is important for the political behavior literature since the two models partly compete to explain the very same phenomena, increasingly widespread throughout the democratized Western world. It is also the first-time that these two models are tested in the context of a post-communist democracy (and in parallel) as the literature largely focuses on the U.S., and, to a lesser extent, Western Europe.

**Funding:** This research received no external funding.

**Institutional Review Board Statement:** The study was conducted in accordance with the Declaration of Helsinki and approved by the Ethics Committee of the London School of Economics and Political Science (REC ref #22670, July 10, 2021).

**Informed Consent Statement:** Informed consent was obtained from all subjects involved in the study.

**Data Availability Statement:** Not applicable.

**Conflicts of Interest:** The author declares no conflict of interest.

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
