# Peer review of "“Polish People Are Starting to Hate Polish People”—Uncovering Emergent Patterns of Electoral Hostility in Post-Communist Europe"

_societies, doi:10.3390/soc12060176_

Round 1

Reviewer 1 Report

The article is interesting and it definitely falls within the scope of the journal, besides it being an actual issue in Western politics. Also, I would like to congratulate author(s) for the ethical protocols used in this research. In order to be published, I recommend author(s) to make more clear their contribution; I will make some remarks that might help with that:

-Section 1.1 author(s) state(s) the following research questions: How have feelings of electoral hostility emerged and developed among Poland’s first time voters and their families? (...) 1) What constitutes an ‘atmosphere' of electoral hostility ? 2) What emotional and experiential cycles lead to electoral hostility ? 3) How do citizens react to electoral hostility ? 4) To what extent is electoral hostility constructed as a cycle of perpetual attitudinal  and behavioural deterioration ? I would expect a clear link between these questions and Table 1- Presentation of results- Conclusions

-Does section "3. Historical and political context" contribute to your question? How? Author(s) should better integrate all results in the conclusion section

-The focus is on the first time Polish voters? Why?

-Numbers of participants are not consistent throughout the paper (how many participants were there in total?), and methodological section could be more simplified amd clear: "built on the conjunction of two qualitative design components: a series of 13 family focus groups involving 35 participants, and 35 semi-structured individual interviews."; "focus groups and individual interviews of up to 70 participants"; "In addition to family focus groups, I conducted 26 individual interviews."

With the experts it happens the same:

"In addition, four experts were interviewed"; "The insights from this section are based on expert interviews conducted with three academics and researchers specialising in Polish contemporary politics."

- Justify why these 3 (or 4?) experts, and not others.. In the same line, author(s) should better justify the sample of the participants (why those places and not others? Possible biases derived from that selection, etc.)

-Small focus groups, or mini-focus groups, comprise four to six participants; author(s) present focus groups "between two and four family members". Two people are far from being a 'focus group', and the main problems with small groups are that they present a limited range of experiences and interactions. Were there big differences between individual and the 'other interviews'?

-Conclusions: is Figure 2 necessary for your article's contribution? You focus on EH (in your title) and not on AP. 

-I would refrain from using this type of phrases independently in the Conclusion section: 'Key findings are summarised in Figure 5.'

-Considering the research design, this article is at most an exploration of the EH, which is a good contribution. Therefore I would be more cautios when presenting results or contributions of the article: "This paper answers the following research question"; "first, by adapting these theories to a new context in which 189 they remain untested, and second, by using this case study to arbitrate between the two 190 competing models claiming to explain why citizens dislike one another because of how 191 they vote."

-use the reference style provided by the journal

Reviewer 2 Report

The main problems relate to samples

1.    Strokes 74 - 76

This research is built on the conjunction of two qualitative design components: a series of 13 family focus groups involving 35 participants, and 35 semi-structured individual interviews. In addition, four experts were interviewed in order to outline the Polish historical and societal context in which the study takes place.

Q: Mentioned 35 individuals: are the individuals from other families or from the same.

2.    Strokes 495 – 498

In total, I conducted 13 different family focus groups. Family members were either all in the same room, or joining the Zoom call separately. The participants ranged in age from 18 to 65. Conversely, when it came to individual interviews, participants ranged in age from 19 to 62. Overall, participants’ mean age was 30.2.

Q1: It's not clear how many ss took part in the first and the second stage separately. The stroke 77 says about 35 individuals for interviews and 35 for focus groups, stroke 487 says about 26, what is correct? Later in page we can see 36 for interview (page 1003).

Q2: For which sample mean age was 30.2? Because there are three samples: for focus groups, for interview and total.

3.    The authors several times says that qualitative studies do not need representative sample, but to have only 13 families from 2 till 4 is too small to my mind.  

4.    Also I would like to know figures where it’s possible, because without figures it look as article in public newspaper. For example, why not to say the % of people living in Warsaw and other large polish cities and the countryside

5.    I am not sure that it's possible to keep whatching body language seeing only face in camera (ss 537 – 538)

6.    Also it’s not clear why interviews and focus-groups were not only in Polish, but also in English and French/ Looks very strange. I need explanations.

7.    It is interesting to read about political emotions to opposition. So I wounder if the pandemic increased or decreased this aggression?

8.    Also I would suggest to make analysis of media: as TV and Radio and how they determine emotions and feelings.

Round 2

Reviewer 1 Report

I would like to congratulate the author(s) for responding to some of my remarks. However, other points were only dealt partially. 

Point 1:  

The reformulation is clarifying. However, what does the author(s) mean(s) by testing a suggestion? [‘The article tests the suggestion that current electoral tension in Poland follows the patterns of the electoral hostility model rather than affective polarization.’]

Point 2:

How does those phenomenon impact first time voters?  Do they add up to your findings?

Point 3 (related to point 9):

Considering the sample and the research design implemented in their research, can we have a definite answer to the initial research question of this study (to point 6, author(s) respond(s): i.e. that this study was not funded and seeks to be exploratory; “It is worth bearing in mind that research for this study has not benefited from any funding and is solely exploratory.”)? In the paper, author(s) state(s) that 'This paper answers the following research question: How have feelings of electoral hostility emerged and developed among Poland’s first time voters and their families?'

The disclaimer is not enough. We are talking about scientific articles and these type of sentences are incorrect and they could mislead the audience.

Response 6:

I could not check the reference provided by the author(s) where family focus groups of 2 members are used in scientific research. Please provide more details about the use of this tool. Aparently, Bruter and Harrison, 2020 studied a similar topic and it would be useful if they have used 2 members family focus groups.

Author Response

Point 1:  

The reformulation is clarifying. However, what does the author(s) mean(s) by testing a suggestion? [‘The article tests the suggestion that current electoral tension in Poland follows the patterns of the electoral hostility model rather than affective polarization.’]

Response 1: There is no hidden meaning here, and this is a key unifying theoretical claim that the research is testing and which unifies the different theoretical expectations described in greater detail in the first part of the article.

I am simply refraining from using the “hypothesis” language because in my view and that of many other qualitativists, in some way, the very restrictive nature of hypothesis formulation is not always the best match for the narrative nature of qualitative research based on interviews and focus groups. This is by no means an original point and authors such as Meinhof, Wodak, or Duchesne have made similar points in the context of their research and often avoided the language of hypothesis testing for the same reason. I am happy to reformulate to clarify if this is deemed ambiguous, for instance into: “This article claims that current electoral tensions in Poland follow etc”.

Point 2:

How does those phenomenon impact first time voters?  Do they add up to your findings?

Response 2:

My expectation was to find first-time voters’ political attitudes to be less tainted by habit than those of more experienced voters. The hope was to receive a narrative that would be more open on questions surrounding perceptions of intra-social relations between voters and less constrained by cumulative habits.

I found that in the case of Polish first-time voters, references to in-group identity were indeed extremely scarce. Unlike electoral hostility, the affective polarisation model necessarily focuses on partisanship as primary identity, and again, partisanship is not seen as the critical foundation of electoral hatred by Polish first time voters.

Moreover, the electoral hostility model focuses on the importance of mirror perceptions which are prominent in the interviews and focus groups. It also describes a specific cycle of ever-deteriorating emotions, all of which are symptomatically mentioned in Polish first-time voters’ narratives.

Point 3 (related to point 9):

Considering the sample and the research design implemented in their research, can we have a definite answer to the initial research question of this study (to point 6, author(s) respond(s): i.e. that this study was not funded and seeks to be exploratory; “It is worth bearing in mind that research for this study has not benefited from any funding and is solely exploratory.”)? In the paper, author(s) state(s) that 'This paper answers the following research question: How have feelings of electoral hostility emerged and developed among Poland’s first time voters and their families?'

The disclaimer is not enough. We are talking about scientific articles and these type of sentences are incorrect and they could mislead the audience.

Response 3: 

The research question for the project was set before fieldwork started and has not changed. It is as stated above.

The reference to the exploratory nature of the research is simply an acknowledgement that I am conducting the research based on qualitative research, which, by nature, does not rely on fully representative samples and imposes caution when interpreting how generalisable findings are.

The reference to the fact that the study was self-funded rather than externally funded is merely intended to provide some context as to this choice of methodology and the use of a relatively limited sample.

That said, it is not unusual to publish research based on even smaller samples than that in qualitative research and whilst I wanted to acknowledge the need for caution in terms of generalisability and acknowledge that further research using other methodologies and larger samples will be useful in that context, I fully stand by the suggestion that the methodology and diversity of the sample means that the research comes up with some important findings in answer to my research questions which can provide very important data points to colleagues working in the fields of electoral hostility and affective polarisation and participate in the vibrant debate currently taking place in that field.

Point 6:

I could not check the reference provided by the author(s) where family focus groups of 2 members are used in scientific research. Please provide more details about the use of this tool. Aparently, Bruter and Harrison, 2020 studied a similar topic and it would be useful if they have used 2 members family focus groups.

Response 6: 

The question of whether there is an optimal (or indeed minimal/maximal) size of focus groups has been the subject of much debate in the literature. I acknowledge that the reviewer’s preference about a minimal size for focus groups which should exclude dyads and triads is shared by many, however, I am simply following an alternative vision shared by multiple others that focus groups are defined by their reliance on interaction between individuals without any strong argument to impose either minimal or maximal sizes, and that in the context of family focus groups, a group of two is and should be acceptable.

Apart from Bruter and Harrison (already cited) which use family focus groups without size restrictions and indeed starting from 2 people in some cases, one could also refer to O. Nyumba, T., Wilson, K., Derrick, C. J., & Mukherjee, N. (2018). “The use of focus group discussion methodology: Insights from two decades of application in conservation”. Methods in Ecology and evolution9(1), 20-32 who refer to focus groups with a starting size of 2 participants.

One could similarly refer to the thoughts expressed in Morgan, D. 1996. Focus groups as qualitative research. London Sage who states: “Who is to say when a group is too large or too small to be called a focus group […] Rather than generate pointless debates about what is or is not a focus group, I prefer to treat focus groups as a ‘broad umbrella’ or ‘big tent’ that can include many different variations.

Reviewer 2 Report

I like the author's work. But my doubts about the sample size are very significant. I recommend the work, but do not insist on this recommendation.